# Harnessing Biofabrication Strategies to Re-Surface Osteochondral Defects: Repair, Enhance, and Regenerate

**DOI:** 10.3390/biomimetics8020260

**Published:** 2023-06-15

**Authors:** Fabiano Bini, Salvatore D’Alessandro, Andrada Pica, Franco Marinozzi, Gianluca Cidonio

**Affiliations:** 1Department of Mechanical and Aerospace Engineering, Sapienza University, 00148 Rome, Italy; fabiano.bini@uniroma1.it (F.B.);; 2Center for Life Nano- & Neuro-Science (CLN2S), Fondazione Istituto Italiano di Tecnologia, 00161 Rome, Italy; 3Department of Biomedical Sciences, University of Sassari, 07100 Sassari, Italy

**Keywords:** resurfacing, osteochondral defects, tissue engineering, bioprinting

## Abstract

Osteochondral tissue (OC) is a complex and multiphasic system comprising cartilage and subchondral bone. The discrete OC architecture is layered with specific zones characterized by different compositions, morphology, collagen orientation, and chondrocyte phenotypes. To date, the treatment of osteochondral defects (OCD) remains a major clinical challenge due to the low self-regenerative capacity of damaged skeletal tissue, as well as the critical lack of functional tissue substitutes. Current clinical approaches fail to fully regenerate damaged OC recapitulating the zonal structure while granting long-term stability. Thus, the development of new biomimetic treatment strategies for the functional repair of OCDs is urgently needed. Here, we review recent developments in the preclinical investigation of novel functional approaches for the resurfacing of skeletal defects. The most recent studies on preclinical augmentation of OCDs and highlights on novel studies for the in vivo replacement of diseased cartilage are presented.

## 1. Introduction

The osteochondral (OC) unit is a highly organized tissue, including superficially layered articular cartilage with underlying subchondral bone connected by an interface region of calcified cartilage [1]. OC carries out the essential functions of transferring and distributing the mechanical load of the skeletal system during movements. The wear and tear, as well as inflammatory and diseased states, can drastically impair the functionality of OC. Osteochondral defects (OCDs) are often caused by severe cartilage loss with extensive damage arising from a single episode of (i) severe trauma, (ii) repetitive microtrauma, or (iii) pathological conditions such as osteoarthritis (OA) [1] and osteochondritis dissecans [2].

A drastic architectural remodelling of the joint is correlated with the localization of OC defects and the consequential development of a complex cascade of degenerative processes that lead to the impairment of joint functionality. Typically, OC lesions are correlated with painful symptoms such as joint-locking phenomena and abnormal joint function. However, the early diagnosis of OC alterations still represents a challenging issue [3,4].

Clinical treatment has been found ineffective for the active repair of OCDs. Surgical interventions are currently failing to provide the ultimate regeneration technique capable of fully repairing OCDs. Thus, new technological advancement is needed for the functional repair and resurfacing of damaged OC lesions. Tissue engineering and regenerative medicine (TERM) is currently offering a valid solution for the functional and clinical treatment of OCDs. This review aims to unravel new methodologies, approaches, and platforms capable of delivering regenerative stimuli while filling the damaged OC gap.

## 2. Osteochondral Tissue Architecture

Articular cartilage is a highly specialized connective tissue that facilitates the transfer of forces between two opposing skeletal elements with a low friction coefficient, aided by the lubrication of synovial fluid and acting as an absorber of weight during sustained static loading [5,6,7]. Chondrocytes are cartilage-specific cells responsible for the organization and maintenance of extracellular matrix (ECM) components. Typically, chondrocytes account for a small percentage of the total volume (1–2%), while the ECM occupy the majority of the cartilage space with 80% water and 20% solid components (i.e., type II collagen, proteoglycan, and other non-collagenous proteins) [3,6,8].

The structure and composition of the articular cartilage vary, moving away from the joint surface and leading to the formation of four regions, namely (i) superficial, (ii) middle, (iii) deep, and (iv) calcified cartilage. Moving from the surface to the depth of the articular cartilage, the concentration of proteoglycan aggregates increases while the water content and the density of chondrocytes decrease. However, the amount of collagen fibrils remains constant throughout the entire cross-section [8]. Moreover, the mechanical strength of each layer gradually increases from the superficial to the calcified zones. most tissues, articular cartilage lacks vascular network, drastically impairing innate regenerative abilities [9]. Hence, even superficial cartilage defects fail to heal and progressively deteriorate to form large lesions, generally correlated with OA onset [3,5,10].

Severe OCDs extend beyond the damaged cartilage, affecting the underlying mineralised bone tissue. The subchondral bone is separated from the calcified cartilage by means of the cement line. It is composed of a thin lamella of cortical bone and the subchondral trabecular bone [6,8]. The subchondral bone is highly vascularized and transports nutrients, water, and waste. Bone is a highly dynamic tissue composed of organic (mainly collagen nanofibers) and inorganic apatite minerals disposed of in a hierarchical structure ranging from the nano- to the macro-scale [11,12,13,14]. Mechanical forces directly influence the remodelling process of bone and, in the case of osteochondral diseases, microstructural changes of the subchondral skeletal tissue [15,16,17].

Due to the poor regenerative capacity of the OC system, current approaches for the treatment of cartilage-bone lesions are only partially effective—aiming to reduce pain and improve joint function while failing to achieve a complete and functional repair. This review aims to describe the current trends and most recent studies of OC pre-clinical approaches, highlighting the innovative techniques in the framework of bone resurfacing.

## 3. Osteochondral Diseases and Clinical Management

The repeated application of a force or the occurrence of severe trauma can cause mechanical disruption of the OC compartments, including fissures, chondral flaps or tears, or loss of a segment of articular cartilage [3].

Alterations of the articular cartilage are mainly correlated with aging, leading to a decrease in articular stiffness and strength. In fact, the age-related chondrocyte apoptosis and the concurrent decrease of water content and proteoglycan size lead to a greater risk of cartilage damage [3,18]. Moreover, anti-inflammatory treatments, diabetes, and hormonal changes due to menopause affect cartilage structure and its mechanical properties, leading to high susceptibility to damage.

A further player in OCDs is OA, a highly debilitating joint-associated dysfunction with progressive cartilage degeneration. OA is typically characterized by synovial fibrosis, an increase in subchondral cortical bone thickness, a decrease in subchondral trabecular bone mass, and the formation of osteophytes cysts (geodes) [1]. During OA progression, alterations in the structure of bones may precede severe degeneration of the cartilage tissue [1,15,17]. The initial stages of OCDs are routinely treated with physiotherapy and drugs (e.g., glucosamine) that can help to relieve pain and improve the function of tissues with small defects. Currently, severe OCDs are treated surgically with either (i) palliative, (ii) reparative, or (iii) regenerative (restorative) treatments.

### 3.1. Palliative Treatments

Palliative treatments (e.g., arthroscopic debridement and lavage) are designed to reduce the articular pain and the risk of osteonecrosis, as well as to slow the progression of the disease. These interventions aim to remove OC fragments, debris, or necrotic material to anticipate possible severe consequences of OCD development. However, the ultimate efficiency of palliative treatments is still controversial and unsuccessful in fully resolving OCDs regeneration [2,19].

### 3.2. Reparative Treatments

Reparative approaches aim to stimulate the subchondral bone to deliver stromal cells (namely, bone marrow stromal cells (BMSCs)) and growth factors (GFs) to the chondral surface to activate tissue healing [20].

Bone marrow stimulating techniques (e.g., arthroscopic abrasion arthroplasty, Pridie drilling, and microfracture) have been found efficacious in approximately 60–70% of young patients with small OC defects (size defect minor than 4 cm^2^) [5]. However, the main shortcoming of these procedures is that the regenerated tissue is composed mostly of fibrocartilage which has reduced functionality and does not duplicate the properties of normal articular cartilage.

Alternatively, other reparative approaches attempt to implant natural or artificial biocompatible acellular material fillers to reinforce or replace portions of damaged joints. The implantation of OC autografts (mosaicplasty) or allografts has been found to efficiently restore the functionality of the damaged joint. However, donor site morbidity, risk of disease transmission, and immune reactions can occur, limiting the use of this technique [3,21,22]. Furthermore, another crucial issue related to OC grafting is the limited availability of grafts that can be achieved without violating the loading joint zone of the articulation.

### 3.3. Regenerative Treatments

Regenerative treatments propose to replace the damaged tissue using a combination of advanced technological platforms that lead to the development of effective yet experimental therapeutic approaches. TERM approaches offer a promising alternative to autologous osteochondral grafts with the combination of biocompatible materials possessing widely varying physical properties with the multilineage differentiation potential of BMSCs [23,24]. Currently, available treatments fail to provide resolutive treatment of OCDs, justifying the need for new functionally regenerative approaches.

## 4. Resurfacing in Orthopedics

Resurfacing of OCDs is an alternative approach to total joint replacement, offering a minimally invasive clinical procedure for the repair of cartilage-bone damages (Figure 1A). The clinical approach involves the sculpting of the underlying bone tissue to accept a resurfacing cup without the presence of an intramedullary implant, thus preserving a more physiological transmission of forces across the joint [25]. This clinical intervention considers minimal bone removal and attempts to maintain the physiological anatomy and biomechanics of the joint in order to reduce the likelihood of revision procedures. Overall, resurfacing procedures are characterized by lower dislocation rates and superior functional outcomes and are thus more suitable for younger active patients compared to conventional total hip arthroplasty [26]. To date, resurfacing arthroplasty has been reported in multiple joints including hips [25], knees [27], and shoulders [27,28].

Modern hip resurfacing approaches comprise the implantation of a large metal-on-metal articulation, cementless acetabular fixation, and cemented femoral fixation [29]. The resurfacing arthroplasty of the hip joint has been observed to offer a more accurate restoration of native hip biomechanics, decrease proximal femoral stress shielding, increase stability, lower incidence of limb length discrepancy, preserve proximal femoral bone stock, and increase range of motion, ease of revision, and rate of return to high demand activities [30,31].

A major concern of hip arthroplasty is related to the generation of metal ion debris from metal-on-metal implants that can lead to local adverse reactions and/or systemic toxicity in some patients. To date, there are no studies comparing resurfacing procedures with standard-bearing total hip arthroplasty on this topic [32]. Nonetheless, recent advances have focused on new alternative bearing couples for hip resurfacing, for instance, the use of ceramic-on-ceramic or metal-on-polyethylene articulation [26].

In shoulder-related pathologies, procedures consisting of biologic glenoid resurfacing with a soft covering and humeral head replacement have also been performed [33].

Different glenoid resurfacing grafts are reported in the literature: lateral meniscus allografts, human acellular dermal matrix, Achilles tendon allografts, shoulder joint capsules, and fascia lata autografts. A common feature of these procedures is the combination of a metallic replacement of the humeral head [33]. Resurfacing the degenerated glenoid with biological material has the goal of producing a durable and biologically active bearing surface that would preserve motion, provide pain relief, and improve joint function [34].

Conversely, resurfacing interventions of the knee joint report several drawbacks, such as (i) greater risk of patellar fracture, (ii) dislocation, (iii) implant failure, (iv) patellar tendon injury, and (v) patellar implant failure. These drawbacks can greatly surpass benefits (e.g., cost-effectiveness, lower number of reoperations, and less anterior knee pain). The lack of evidence of a global benefit for resurfacing approaches in comparison with total tissue replacement is preventing a rapid translation of new resurfacing technologies. However, the ease of fabrication, implantation, and follow-up are currently encouraging a clinical application of novel TERM OCD resurfacing techniques [34] (Figure 1B).

## 5. Technologies for Bone Resurfacing

Novel technological advancements are coming to the fore as functional platforms for OCD resurfacing and treatment. A number of recent studies have reported the use of novel scaffolding materials and methodologies for the active repair of cartilage and underlying bone, comprising cutting-edge in situ fabrication and implantation of resurfacing constructs (Figure 1C). Herein, we highlight the most relevant engineering methods capable of fabricating scaffolds that offer the potential to locally repair and enhance the surface of OCDs.

### 5.1. Electrospinning

Electrospinning is an electrohydrodynamic technique based on the fabrication of nanostructures. A polymeric material can be extruded from a spinneret and directed towards a target (collector) guided by an electric field. During the extrusion process, the solvent evaporates, drying the fibrous material deposited onto the collector and leaving a woven lattice structure that closely resembles the fiber dimensions and random arrangements of the extracellular matrix (ECM) [35].

The morphological properties of the scaffolds can be adjusted by tuning experimental parameters such as (i) solution (polymer concentration and solvent used) [36,37], (ii) voltage [35,38], (iii) flow rate [39], (iv) collector-to-nozzle distance [39], and (v) environmental parameters (e.g., temperature and humidity) [40]. Both the collector material and topological features can be employed to harness randomly arranged or aligned fibers, drastically augmenting the complexity of the deposited scaffold [39]. Moreover, the chemical composition and associated properties of the scaffold can be tuned using a plethora of natural (e.g., silk fibroin, gelatine, chitosan, etc.) or synthetic (e.g., polycaprolactone (PCL), poly-lactic acid (PLA), polyethylene glycol (PEG)) materials [40,41]. Indeed, the combination of more than one type of material has been proven to be optimal for the fabrication of complex hierarchical scaffolds [42,43].

Furthermore, complex fiber geometry, such as core-shell, can be produced using a coaxial or multiaxial spinneret [44], introducing further complexity to the system with additional correlated functional properties. In order to enhance superficial properties, post-spinning treatments, such as plasma treatment [45], ion sputtering [46], oxidation [47], and corona discharge [48], can be used to enhance cellular adhesion, proliferation, and differentiation.

### 5.2. 3D Bioprinting

Three-dimensional bioprinting aims to pattern and assemble, layer-by-layer, bioinks (cells and biomaterials) with functional tissue-specific properties to produce bio-engineered structures. This 3D assembling technology allows for the fabrication of three-dimensional scaffolds with a pre-programmed structure, offering the possibility of patterning both biomaterials and living cells to closely resemble tissue architecture [49]. A number of 3D bioprinting platforms are currently available for patterning functional scaffolds, namely inkjet–, laser–, and extrusion–based bioprinting.

#### 5.2.1. Inkjet–Based Bioprinting

Scaffolds are assembled drop-by-drop with inkjet-based bioprinting [50] using piezoelectric [51], thermal [52], and electrostatic inkjet printheads [53]. Thermal or mechanical stress represents a significant limitation that can cause extensive damage to the extruded cells. Moreover, although the inkjet-based bioprinting technique is still an easy, fast, and versatile method, it cannot manage bio-inks with high cell density since they would have significant disadvantages with cell death and nozzle blockage during printing [54].

#### 5.2.2. Laser–Assisted Bioprinting

Laser-assisted 3D bioprinting offers high performance and resolution for the deposition and patterning of bio-inks [55,56,57,58,59]. This method is nozzle-free and aims to pattern–seed materials from a donor site to a collector slide using laser impulses. The donor slide is covered with a layer that can absorb the radiation energy that, in turn, causes a precise ejection of the cells. Therefore, laser-based bioprinting techniques can produce precise constructs with low cell damage but require specific cell bio-ink mechanical properties and expensive production costs [54].

#### 5.2.3. Extrusion–Based Bioprinting

Extrusion-based bioprinting is among the most widely exploited fabrication technologies for TERM purposes [60]. The printing ability of extrusion-based bioprinting depends on the density of the encapsulated cells and on the type of material ink (e.g., hydrogels, micro-carriers, tissue spheroids, cell pellet, or decellularized matrix components) as well as on the ultimate crosslinking approach, comprising exposure time and intensity [60]. Nevertheless, extrusion–based bioprinting can accept a large variety of printable materials, greatly widening the plethora of possible applications with this approach. Indeed, extrusion–based bioprinting represents an inexpensive and versatile method to produce scaffolds, modulating the printer parameters and experimental setup, but it remains limited in terms of resolution (still above 100 µm) [58,59]. Considering the flexibility and capability to print scaffolds with a controlled porous structure using a simple setup to extrude the bio-ink, studies used the 3D printing manipulators as a tool for extrusion–based bioprinting to treat the defect in vivo [61,62]. Recent advancements foster the development of in situ 3D bioprinting applications (Figure 1C). This revolutionizing approach aims to directly deliver and pattern the bio-ink within the site of interest through handheld, portable, or robotically-assisted 3D-bioprinting platforms. The in situ approach might be able to repair defects in a short time, replicating and matching complex anatomical shapes that require new and reparative engineering [58].

## 6. New Platforms for Functional Tissue Resurfacing

Advancements in TERM technologies are paving the way for engineering a functional solution for OCD treatment. Particularly, novel platforms such as 3D bioprinting that have demonstrated the ability to generate clinically relevant resurfacing options are of great interest for the biomimetic regeneration of OC tissue (Figure 2A).

### 6.1. Electrospinning to Re-Engineer Osteochondral Surfaces

A number of recent studies highlighted that electrospinning is particularly suitable for the resurfacing of damaged cartilage tissue [62,66]. However, due to the inherent complexity of the OC region, the development of a hierarchical scaffold still represents a challenge. Indeed, cartilage is partially anisotropic with layered, aligned, and parallel collagen fibers. Electrospun fibers can be directed and collected in an orderly fashion, producing viable scaffolds for a number of applications [40,43,67,68]. In particular, constructs fabricated by harnessing electrospinning technologies have been found to support cell viability and proliferation, fostering homogeneous colonization of the graft [62]. However, the integration of scaffolds with both native cartilage and bone tissues remains a major challenge. The implant must match a specific degradation rate to allow the targeted replacement of the scaffold with new cartilaginous tissue while allowing seeded cells to integrate with the underlying bone. The functionality of the implant can be enhanced by the targeted release of chemotactic factors from the degradable polymeric mesh. To this purpose, insulin growth factor (IGF-1) has been recently incorporated in a biodegradable electrospun matrix of polylactide-co-glycolide (PLGA) and PCL to enhance the formation of cartilage across graft-host in vivo [67].

Recent studies have introduced new methodologies to improve scaffold cell colonization producing aligned microfibers scaffold that exhibit a higher level of cell viability than random microfiber arrangements. Gluais and co-workers [68] fabricated a cell-free biodegradable PCL scaffold to repair a superficially damaged annulus fibrosus (AF) of an ovine model. Circular scaffolds (1.5 cm in diameter) were coated with fetal bovine serum (FBS) overnight in an incubator at 37 °C and 5% CO_2_. After four weeks of implants, histological and immunohistochemical analyses revealed the integration with the adjacent tissue of the biomimetic electrospun implant, successfully demonstrating a cell-free approach to mimic the native cartilage mechanical properties, promoting spontaneous cell colonization, proliferation, and organization. The aligned fiber architecture was found to improve inductive behaviour, granting high viability of the infiltrating cells, depositing fibrous collagen around the graft, and increasing the attachment of the implant with the cartilage tissue.

A further study by Ren et al. [63] reported aligned porous PLLA electrospun fibrous membrane with a coating increasing its superficial biomimetics properties designed to repair articular cartilage defects (Figure 2B). The scaffold was coated with chondroitin sulfate (CS) using polydopamine (PDA) as an adhesive polymeric bridge and functionalized with rabbit BMSCs. The in vivo study demonstrates that the coated scaffold with PDA/CS facilitated the attachment to the cartilage defect and increased the generation of cartilage in the defect site. However, the aforementioned approach is still not sufficient to provide a stable and functional resurfacing solution.

Indeed, cartilage is a zonal tissue with specific mechanical properties and ECM composition. Articular cartilage is mechanically stable, thanks to a thin film of synovium that greatly reduces friction. Nevertheless, the wearability of this tissue is enhanced when the synovium is limited, and the constant mechanical contact with underlying bone tissue may lead to disruption, extensive pain, and discomfort. To provide a biomimetic zonal solution (Figure 2C), Steele and collaborators [64] recently engineered a novel scaffold comprising a gradient of stiffness mimicking the hierarchical architecture of the articular cartilage using PCL and a combination of multiple fabrication strategies: electrospinning, spherical porogen leaching [69], directional freezing [70], and melt–electrowriting (MEW) [71]. An in vivo porcine model comprising a double OC lesion in the trochlear groove was employed. Results demonstrated that the micro-architecture of the implanted scaffolds was found stable and unaltered following six months of treatment of OCD of the large animal model. This method offers a noticeable solution to treat OC defects with a scaffold that mimics the native tissue properties reproducing a similar hierarchical native architecture. However, the engineering of this zonal micro-structured scaffold makes use of multiple fabrication strategies that involve a complex and highly time-consuming production procedure not suitable for scale-up production.

### 6.2. 3D Bioprinting Approaches for Bone Resurfacing

The ability to mimic a hierarchical anatomical structure is crucial to avoid the possibility of engineering a construct that drives incomplete regeneration of the native tissue [40,41]. Conventional bulk scaffolds are not suitable for reproducing the anatomical conformation of the joint and are limited in replicating the specific shape of an OC defect [72,73]. Thus, advanced 3D bioprinting technology is emerging for the fabrication of functional constructs to restore the articular defect by regenerating the native tissue structure [74,75,76,77,78].

Significant effort has been invested in developing repair scaffolds for mimicking the structural, morphological, chemical, and cellular gradients of native articular cartilage and subchondral bone [79]. The tissue architecture is characterized by different fibrous collagen, which is the most adopted biomaterial for tissue-engineered structure. Collagen is a structural protein that ensures mechanical properties [80] and forms the basis of connective tissue [81]. For these reasons, the literature has studied the application of collagen hydrogel in different concentrations as high as 1.75 [82], 2 [83], and 2.4% [84] to reproduce the mechanical properties of the native tissue. These collagen concentrations are insufficient to restore the tissue characteristic and remain quite fragile. Thus, in a recent study, Beketov et al. [85] studied the applicability of a bio-ink based on 4% collagen for the in vivo cartilage formation, obtaining a tissue substitute rich in type II collagen with higher mechanical properties compared to low conventional concentrations.

Another relevant TERM challenge is the lack of adequate vascularization to support the growth and viability of new tissues that require blood supply [86]. Significant and recent studies are increasing the knowledge of the angiogenic process to stimulate the generation of a vessel network for nutrient and gas transport. Indeed, limited vascularization involves a decrease in the rate of cell proliferation [87]. A number of research groups have attempted to address this problem by engineering scaffolds with pores capable of stimulating an increase in gas and liquid perfusion, ultimately improving tissue growth [71,72,88]. Recently, Sun et al. [65] created an anisotropic pore gradient-structured cartilage 3D scaffold combining the printing of hydrogel and PCL fibers with BMSC and hypoxia-inducible factor 1 α (HIF1α). The gradient-structured cartilage construct was found to show higher cartilage repair effect in vitro and in vivo with respect to traditional scaffold due to improved cell proliferation and neo-angiogenesis, guided by HIF1-α- mediated focal adhesion kinase (FAK) axis activation (Figure 2D). The approaches reported here hold great potential for clinical translation. However, the possibility of using 3D bioprinting for the engineering of functional scaffolds for clinical repair is currently limited by the time required for the expansion of bioprinted cells and the overall maturation needed for the 3D-printed constructs to be implanted in vivo.

Typically, the maturation process can be accomplished between two and four weeks, greatly limiting rapid use in a clinical scenario. Ideally, the direct 3D printing of a viable and functional implant in situ would immediately help the patient and solve a number of challenges in the clinical theatre.

### 6.3. Advance Technology In-Vivo via Tissue Engineering Osteo—Chondral Resurfacing

Current clinical strategies adopted to manage OCDs, relieve pain and re-establish articular movement [36] but cannot fully restore the entire OC architecture and functionality [37]. Thus, the development of new treatment strategies is essential, comprising safe and effective delivery methods for the implantation of OCDs resurfacing implants in vivo. The latest developments in pre-clinical investigation of more effective methods with a direct fabrication/implantation approach for resurfacing skeletal defects (Figure 3A) are listed here. In Table 1, we report recent studies targeting pre-clinical augmentation of OCDs and highlighting novel work on in vivo cartilage resurfacing applications.

The clinical use of a scaffold for OC repair requires the engineering and manufacturing of the graft, followed by packaging and distributing before ultimate clinical application. A versatile alternative would comprise the application of electrospinning in situ for the re-surfacing of damaged tissues. This alternative has been recently proposed with a direct application on superficial wounding for the fabrication of personalized wound dressing [90,91].

In situ, the deposition of nanofibers is implemented by a portable, safe, and easy-to-use device that constantly spins fibers onto the object. Currently, in situ electrospinning of nanofibres is not yet suitable for an OC defect because of the complexity of the anatomical district and the ultimate difficulty in depositing nanofibers in a specific and small portion of damaged tissue. Moreover, the electrospinning apparatus often involves the use of toxic solvents with potentially compromising effects on the biocompatibility of the implant. Thus, the development of a portable melt electrospinning apparatus to avoid toxic solvents and to improve deposition control may represent a breakthrough advancement of the technology related to in situ electrospun nanofibers deposition. However, the elevated electrical discharge needed for this technology to deposit sub-micron size fibers greatly limits the tremendous potential of MEW apparatus for in situ OCD resurfacing.

Thus, researchers are studying new techniques and methods for improving surgical tissue engineering procedures using robotic-assisted 3D bioprinting for the functional repair of OCDs. A recent study [89] demonstrated that cartilage injury could be treated by using robotic-assisted in situ 3D bioprinting technology restoring an in vivo rabbit OCD with hyaluronic acid methacrylate (HAMA) and acrylate-terminated four-armed polyethylene glycol bio-ink (Figure 3B). This technology is highly appropriate for improving surgical procedures, demonstrating that it is a viable alternative for skeletal defect restoration and eliminates the need for in vitro scaffold preparation, reducing the risk of contamination and treatment time.

Li and co-workers [92] recently described the in situ robotically-assisted repair of a segmental skeletal defect (Figure 3C). A critical bone defect was repaired using a novel bio-ink formulation comprising partially-crosslinked alginate mixed with gelatin methacryloyl (GelMA) and poly(ethylene glycol) diacrylate (PEGDA). A post-printing irradiation with UV at 150 mW/cm^2^ for less than 10 s was found to preserve high cell viability and functionality. Following 12 weeks of in situ printing, more than 70% of the defect was found to be filled with a high bone volume fraction, demonstrating the outstanding potential for revolutionizing the clinical intervention in OCD repair cases.

In particular, in a recent study by Lipskas and co-workers [93], a robotic-assisted, minimally invasive surgery (MIS) and 3D-bioprinting processes to restore an OC defect were presented. Results from the in situ remote–center–of–motion–(RCM) guided deposition in an ovine OCD model were found promising following the resurfacing using a novel alginate-PEGDA system. The accuracy of the system allowed the precise clinical intervention with a spatial resolution of 0.06 ± 0.14 mm, thereby demonstrating a viable in vivo option for repairing OCDs.
biomimetics-08-00260-t001_Table 1Table 1TERM approaches for clinically-relevant OCD resurfacing.
AnimalTherapy (T) and Findings (F)Ref.Electrospinning ResurfacingCalfTA cartilage graft (PLGA/PCL) enhanced with chemotactic factor (IGF-1) [67]FThe defect regeneration is improved by promoting cell-mediated integrative cartilage.OvineTA cell-free PCL electrospun scaffold made with aligned microfibers [68]FThe aligned scaffold exhibited high levels of cell colonization, demonstrating that the aligned fibers improve cell viability.RabbitTAn aligned porous (PLLA) electrospun-coated scaffold [63]FThe biological effect is significantly increased, and the combination of aligned porous hierarchical structure exhibits high regenerative properties.PorcineTA hierarchical scaffold designed to mimic the articular cartilage structure (multiple techniques)[64]FThe retention, osteointegration, and prolonged degradation of the scaffold were acceptable with beneficial effects.3D Bioprinting ResurfacingRatTA microfluidic extruder to compartmentalize OCD[94]FThe possibility of mimicking the biological and mechanical gradient structure of cartilage interface is demonstrated.RatTA construct with collagenous bio-ink for cartilage regeneration[85]FA high concentration of collagen generates new tissue rich in GAGs and type II collagen.RabbitTAn anisotropic pore gradient-structured cartilage 3D scaffold combining printing of hydrogel and PCL fibers with BMSC and HIF1α/FAK. [65]FThe scaffold generated and maintained stable cartilage phenotype in different layers, and the ECM implant composition induced cartilage similar to native tissue.In situapproachesRabbitTA robotic arm is used for in situ 3D printing process, depositing the bio-ink directly inside the defect.[89]FThe regenerated tissue faithfully reproduces the native tissue composition and morphology, demonstrating that the technology can improve the surgical procedure in clinical application.


## 7. Summary, Challenges, and Future Perspectives

Over the past few decades, TERM strategies in the regeneration of skeletal tissues have advanced considerably, taking into account the promising results obtained from in vivo experiments. The interest of the scientific community in designing new reparative strategies is justified by the lack of regenerative treatments for OCDs. It is worth noticing that, to date, the traditional clinical approaches are mainly palliative [95], including the washing or the debridement of the defect to relieve the symptoms without healing the articular surface. Reparative techniques are used to either stabilize the defect or generate fibrocartilage (drilling, arthroscopic-assisted fixation, microfracture) while failing to ultimately provide a solution for the repair of OCDs. All of these traditional approaches have limitations in reporting variable and unpredictable outcomes [96].

TERM offers a promising alternative strategy for treating skeletal injuries, restoring the tissue district completely. The ability to repair or regenerate cartilage could have an extraordinary impact on the treatment of OCD, improving the lifestyle of the aging population. To this end, in this review, we emphasized new TERM in vivo treatment procedures as reliable pre-clinical proof of concept for OCD repair. To achieve tangible and clinically relevant results, we have reported in vivo studies that investigate the current situation of OCD repair.

Cartilage tissue engineering based on electrospinning technology may represent a promising yet limited strategy to offer a viable solution to OCD. The electrospun nanofibers have received much attention thanks to their ability to create structures with a microstructure similar to ECM [96,97], high surface-to-volume ratio, and interconnected, three-dimensional porous architecture [98,99]. Nevertheless, electrospun scaffolds are still incapable of reproducing the complex superficial morphology of cartilaginous native tissue. However, the use of electrospinning has been found to promote chondrocyte alignment and migration using an aligned arrangement of the fibers. In addition, a recent approach involves the superficial functionalization of the scaffold by coating the nano-fibers with bioactive molecules of interest, which has been found to promote cytocompatibility, cell adhesion, and proliferation.

Nonetheless, the implants present great limitations in mimicking the hierarchical OC tissue since, with the current electrospinning technology, it is difficult to produce a gradient-structure scaffold. Instead, the 3D bioprinting technology is a promising tool for the fabrication of predetermined architecture due to the layer-by-layer deposition of bio-inks that gives the opportunity for precise spatial control on the deposition with different chemical and biological compositions suitable to produce gradient scaffolds. The differentiation of the lattice structure composition contributes to the heterogeneous differentiation of stem cells within the construct. Thus, the combination of different bio-ink with extrusion-based bioprinting allows the regeneration of heterogeneous tissues, such as the OC interface. However, there are several challenges, including the integration of adjacent tissues, the reproduction of biochemical properties, and the efficient combination of stem cells and signalling factors, yet to be solved. Future investigations will need to address these limitations of 3D bioprinting, creating scaffolds with higher performance and achieving real usage in clinical treatments of osteochondral defects. Microfluidic 3D bioprinting [100] holds tremendous potential for the rapid fabrication of hierarchical multi-tissue structures but is still far from reaching the majority of orthopaedic theatres.

Although these methods are promising, clinical approaches are commonly used in vivo for healing small focal defects. If the defect is large enough (>4 cm), it is necessary to reproduce the shape of the native anatomic morphological region [101]. In fact, restoring the superficial zone of cartilage is mandatory since it contributes to load distribution during joint movement, therefore playing a vital role in the maintenance of cartilage function.

In this context, recent works using an in situ robotic-assisted 3D bio-printing depositing bio-ink for the repair of OCDs are at the forefront of clinical innovation. Indeed, the in situ deposition of a therapeutic implant holds great potential for the patient-specific repair of musculoskeletal disorders and damages. Thus, the approach highlighted here could allow direct treatment planning and application on the patient. Scanning the site can record the defect surface and generate a model of the defect volume with specific software. Therefore, in situ 3D bio-printing technology can improve the surgical procedure for OC injury and increase the morphology graft accuracy, reducing the damage to the joint.

During the past decades, the development of TE approaches in the regeneration of cartilage and OC tissue has had a considerable evolution, given a notable impact on pre-clinical studies. Improvement in the understanding of aligned fibers, coating, and stratified architecture of OC tissue has led researchers to test the constructs in vivo with promising results.

Despite several challenges, the rapid evolution of TE technologies into novel in situ bioprinting approaches, combined with the combined effort of bioengineers, biologists, and clinicians, will allow us to take a step closer to OCD repair and, ultimately, reduce the socioeconomic burden of musculoskeletal joint disease.

## Figures and Tables

**Figure 1 biomimetics-08-00260-f001:**
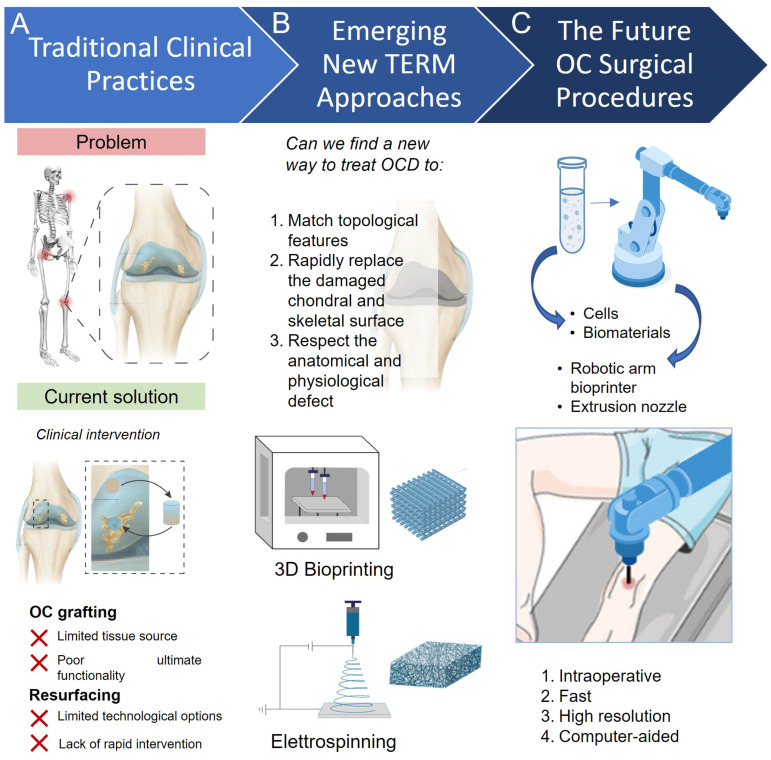
Traditional and emerging strategies for the treatment of osteochondral diseases. (**A**) Typical rapid inflammation and deterioration of OCD may enhance the damage of both cartilage and bone tissue. Current clinical intervention uses osteochondral grafting to rapidly resurface damaged OCD and treat diseased joints. However, the poor availability of autologous tissue associated with the painful and repetitive intervention is opening for (**B**) new TERM strategies (e.g., 3D bioprinting and electrospinning), which promise the rapid replacement of damaged OCD tissues. Particularly, robotic-assisted 3D bioprinting (**C**) has been found capable of high-resolution resurfacing within in situ structures to repair OC tissue and defects.

**Figure 2 biomimetics-08-00260-f002:**
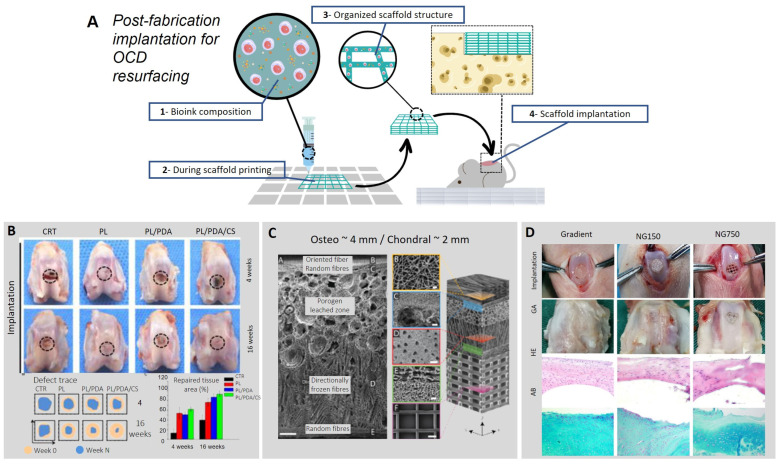
Biofabrication of an implantable construct for OCD repair. (**A**) Schematization of bio-ink printing technology. The 3D bioprinting of cells and biomaterials comprises a step of cell encapsulation within biomaterials, creating a bio-ink with specific viscoelastic properties to guide 3D deposition. Scaffolds can then be fabricated with a specific shape and fiber morphology, easily adaptable to the defect size. Following scaffold design and fabrication, the graft can be implanted within the defect. (**B**) Digital photographs of the defect with the four different approaches (empty defect, PL, PL/PDA/CS) after 4 and 16 weeks. PL is the composition of the scaffold, which is made with Poly (L-lactic acid) (PLLA). PDA is a polydopamine coating. The light brown area indicates the defect area at day 0, and the blue area indicates the defect area after 4 and 16 weeks. Adapted with permissions from [63]. (**C**) Micro-structured scaffold for OCD treatment is fabricated by harnessing a multi-platform approach, demonstrating the ability to mimic the hierarchical composition of the native tissue. Each section reproduces the mechanical and architectural features of the layered OC tissue. The superficial layer is fabricated using melt-electrowriting (MEW) technology. Then, a frozen-electrospun layer with random fibers is interfaced with the superficial layer, offering anchoring points for the underlying section fabricated with directionally frozen foam. Furthermore, a porogen-electrospun interface precedes the last layer composed of aligned electrospun fibers. Immunohistochemistry investigation following culturing demonstrated the extensive expression of collagen type I and type II. Adapted with permissions from [64]. (**D**) The gradient-structured scaffold was compared with different scaffolds, NG150 (pore size = 150 μm) and NG750 (pore size = 750 μm). A gross appearance (GA) analysis of the repaired cartilage was conducted after 24 weeks. The section was stained with HE and AB to indicate the presence of proteoglycans, improved cell filling and morphology, and greater angiogenesis in the Gradient group. Adapted with permissions from [65].

**Figure 3 biomimetics-08-00260-f003:**
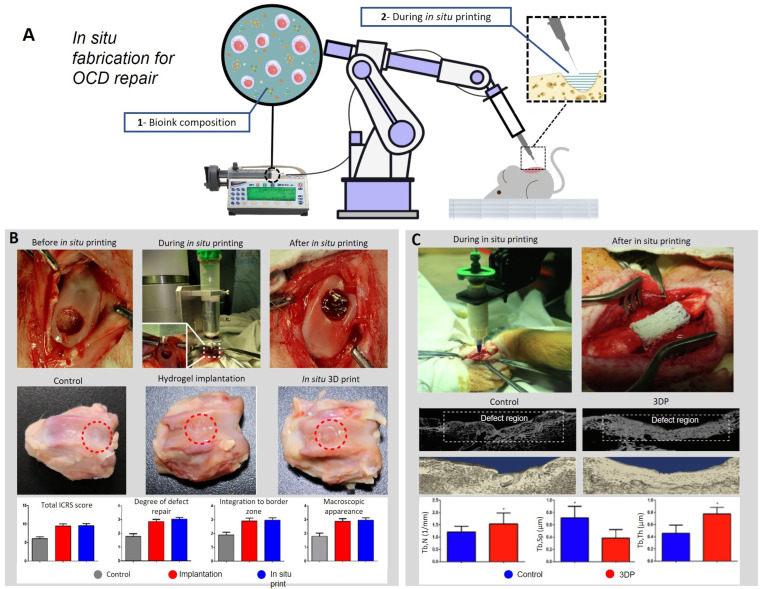
(**A**) Schematization of bio-ink in situ printing technology. The bio-ink is made with specific viscoelastic properties to allow the deposition during in situ printing. The scaffold is directly printed inside the defect, reproducing the native shape of the tissue. (**B**) In situ bioprinting process applied on a rabbit knee joint before, during, and after the treatment. All the samples were harvested after 12 weeks. Digital pictures show the defect after 12 weeks of the control, the hydrogel scaffold implantation, and the in-situ 3D printed scaffold. The grafts show details about the results. Adapted with permission from [89]. (**C**) These pictures show the whole process of in-situ 3D bio-printing. Micro CT scans were conducted 12 weeks post-surgery. The grafts show details about the bone volume fraction (BV/TV). Mean ± SD, * *p* < 0.05. Adapted with permission from [9].

## Data Availability

No data has been generated in writing this review.

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
