# Peer review of "Harnessing Biofabrication Strategies to Re-Surface Osteochondral Defects: Repair, Enhance, and Regenerate"

_biomimetics, 2023, doi:10.3390/biomimetics8020260_

Round 1
Reviewer 1 Report
The submitted manuscript is devoted to an actual topic regeneration of osteo-chondral defects, where the authors pay special attention to tissue engineering methods, such as electrospinning and 3D bioprinting. The review includes the latest relevant information in this area, the most of the cited sources were published no more than 10 years ago. In general, the review is very interesting, but requires minor revision. There were several comments on the style of narration and presentation of the material.
Comments to the text:
In lines 30 and 89, the authors repeat the abbreviation of the term "osteoarthritis", and in lines 45 and 135, the authors repeat the abbreviation of the term "Tissue engineering and regenerative medicine (TERM)". There are no transcripts of the abbreviations TE and ES.
The chapters "Palliative treatment" and "Reparative treatments" have the same numbering. It is necessary to correct the numbering in chapter "5.2.3. D Bioprinting".
The term "Skeletal Resurfacing" is too generalized and not adequate enough, perhaps it should be replaced with the term "Bone Resurfacing "
Author Response
We thank the reviewer for their constructive comments that have greatly improved the manuscript. To address comments to the text we have included/excluded portions in the manuscript using the blue color – and reported the amended selections as follows.
- We have now amended the acronyms OA and TERM in the manuscript. To avoid confusion and mislead the reader, we have amended the TE acronym with the full "TERM" abbreviation, and replaced the ES acronym with the full name electrospinning.
- We thank the reviewer for highlighting these typos. We have now amended the numbering.
- We would like to thank the reviewer for this suggestion. We have now implemented the suggestion in the text by replacing “skeletal resurfacing” with the more appropriate “bone resurfacing”.
Reviewer 2 Report
This review article provides a comprehensive overview of the Harnessing Biofabrication strategies to re-surface osteochondral defects. The authors present a detailed description of recent developments in the preclinical investigation of novel functional approaches for the resurfacing of skeletal defects.
However, there are a few areas that require attention in the article. Firstly, the clarity of the images, particularly for the coordinates in Figure 3B and C, needs improvement. The current quality of the pictures hampers the reader's ability to interpret the information effectively.
Secondly, it is important to address the formatting of the references, specifically the 9th and 85th references. The inclusion of numbers after the journal name "Science" in those particular references is perplexing and should be clarified or corrected for consistency and coherence.
Author Response
We would like to thank the reviewer for the precious considerations and comments that have greatly improved the quality of our manuscript.
We have now amended the manuscript by highlighting in blue the corrections suggested by the referee.
We have provided a point-by-point response to the reviewer’s comments as follows:
- As requested, we have now re-shaped Figure 3, augmenting the resolution to help the reader in understanding the most important details that we have reported in this panel.
- We have amended references 9 and 85 to match the journal format. We can now confirm that all the references are formatted as requested by the journal.